# Yield Response of Spring Maize under Future Climate and the Effects of Adaptation Measures in Northeast China

**DOI:** 10.3390/plants11131634

**Published:** 2022-06-21

**Authors:** Jackson K. Koimbori, Shuai Wang, Jie Pan, Liping Guo, Kuo Li

**Affiliations:** 1Key Lab for Agro-Environment, Ministry of Agriculture and Rural Affairs, Institute of Environment and Sustainable Development in Agriculture, Chinese Academy of Agricultural Sciences, Beijing 100081, China; jackson_kinyanjui@yahoo.co.uk (J.K.K.); panjie@caas.cn (J.P.); guoliping@caas.cn (L.G.); 2College of Land and Environment, Shenyang Agricultural University, Shenyang 110866, China; shuaiwang666@syau.edu.cn

**Keywords:** climate change, CERES-Maize model, climate scenarios, maize yields, adaptation measures

## Abstract

Agriculture production has been found to be the most sensitive sector to climate change. Northeast China (NEC) is one of the world’s major regions for spring maize production and it has been affected by climate change due to increases in temperature and decreases in sunshine hours and precipitation levels over the past few decades. In this study, the CERES-Maize model-v4.7 was adopted to assess the impact of future climatic change on the yield of spring maize in NEC and the effect of adaptation measures in two future periods, the 2030s (2021 to 2040) and the 2050s (2041 to 2060) relative to the baseline (1986 to 2005) under RCP4.5 and RCP8.5 scenarios. The results showed that increased temperatures and the decreases in both the precipitation level and sunshine hours in the NEC at six representative sites in the 2030s and 2050s periods based on RCP4.5 and RCP8.5 climate scenarios would shorten the maize growth durations by (1–38 days) and this would result in a reduction in maize yield by (2.5–26.4%). Adaptation measures, including altered planting date, supplemental irrigation and use of cultivars with longer growth periods could offset some negative impacts of yield decrease in maize. For high-temperature-sensitive cultivars, the adoption of early planting, cultivar change and adding irrigation practices could lead to an increase in maize yield by 23.7–43.6% and these measures were shown to be effective adaptation options towards reducing yield loss from climate change. The simulation results exhibited the effective contribution of appropriate adaptation measures in eliminating the negative impact of future climate change on maize yield.

## 1. Introduction

In China, maize is one of the most predominant grain crops grown for purposes of human food, animal feed, and as an industrial raw material [1]. The maize production area in Northeast China (NEC), where the famous Northeast Corn belt is located, accounts for 30% of total maize produced in China, thereby playing a pivotal role in ensuring food security in China [2]. Agriculture has been found to be the most sensitive sector to climate change despite the availability of developed technologies nowadays [3].

According to Yin et al., (2015) [1], in the past century, China’s air temperatures have increased by 0.5 to 0.8 °C and an increase of 2 °C has also been projected to occur in over 60% of all regions in China by the end of the 21st century. Additionally, elevations in atmospheric CO_2_ concentration (eCO2) could affect the carbon and nitrogen metabolism in the crops and soil nutrient cycling in the soil thus affecting soil fertility, leading to fluctuations in maize yield [4,5]. Northeast China (NEC) has been identified as one of the regions most affected by climate change [6], with recorded increases in temperature and decreases in sunshine hours and precipitation levels [7]. Over the past five decades, the effects of climate change have been felt in NEC with a recorded temperature increase of 0.38 °C per decade and this has resulted in a variation in maize production within the region [2]. As a ripple effect due to climate change, spring maize in NEC has recorded changes over the past few decades resulting in the shift of its maize growing area towards the higher latitudes [8]. Additionally, an increase in maize yield in NEC has been attributed to a variation in solar radiation that has been recorded during the maize growing season from May to September [3,9]. In NEC including Liaoning, Jilin and Heilongjiang provinces, yields were predicted to decrease for both rainfed and irrigated maize by 35% in 2080 under A2 scenario and 30% under B2 scenario [10]. The worst impact of climate change exhibited on maize yield among all the Representative Concentration Pathways (RCPs) without incorporating the effects of CO_2_ fertilization occurred under RCP8.5 scenario, with projected future decreases of 14% for the 2020s, 24% for the 2050s and 46% for the 2080s [1] and with another similar future projection decrease of 6.3% for the 2020s, 18.4% for the 2050s and 47.5% for the 2080s [2]. However, the actual yields in NEC showed a tendency to increase by 1.27 metric tons (t) per hectare (ha) per decade from 1961 to 2010 [11]. According to Wang et al., (2011) [12], the average maize yield in NEC is projected to decrease by more than 15% by 2050 based on 90% of the total 120 projected scenarios. Therefore, to evaluate the impact of future climate change on the yield of spring maize in NEC, we used the simulations based on different climate change and crop parameters to estimate future yield and the level of uncertainty using the CERES-Maize model.

Adaptation measures to mitigate the negative effect of climate change have been found to be very critical in ensuring food security [13,14]. According to the IPCC (2007) [13] Fourth Assessment Report on Climate Change and Tao et al. [15], a few adaptation strategies to cope with the future changing climate include adjusting sowing dates, changing cultivars and improving management practices such as irrigation, planting density and fertilizer application [11]. Additionally, the use of hybrid cultivars and longer season cultivars has helped improve the grain filling rate and yield of maize in NEC [16]. The adoption of several management measures as opposed to using a single mitigation measure has also been found to be effective in eliminating the negative impacts of climate change [17]. Therefore, it is important to examine what adaptation measures could be applied to mitigate against the impacts of climate change on spring maize production in NEC.

In this study, we used the CERES-Maize model (DSSAT) v4.7 to assess the impact of future climatic change on the yield of spring maize in Northeast China and the effect of adaptation measures under different parameters for two future periods, the 2030s (2021 to 2040) and the 2050s (2041 to 2060) relative to the baseline (1986 to 2005) under RCP4.5 and RCP8.5 scenarios. The latest RCP4.5 and RCP8.5 scenarios adopted in the Fifth Assessment Report (AR5) of the United Nations Intergovernmental Panel on Climate Change (IPCC) were applied in this chapter.

## 2. Materials and Methods

### 2.1. Study Sites

The study area of Northeast China consists of 3 provinces namely Liaoning, Jilin and Heilongjiang and covers an area of 793,300 km^2^ with an estimated population of about 109 million [2]. NEC lies among the Daxingan, Xiaoxingan and Changbai mountains, located at an elevation lower than 200 m (Figure 1). The region is characterized by a continental monsoon climate with an average air temperature between 15 and 25 °C and annual precipitation ranging between 400 and 650 mm, respectively [2]. The precipitation in NEC mostly occurs in summer and autumn coinciding with the maize growing period from May to September [1]. The terrain in NEC is fairly flat characterized by black and brown loamy fertile soil which has helped improve the drainage and fertility of the area [18]. To drive the CERES-Maize crop modeling software, suitable study sites that contain high-quality data were selected from the Agricultural Meteorological Experimental Stations (AMESs) as illustrated in Figure 1 and to enhance the performance of the CERES-Maize (DSSAT) V4.7 crop model, the suitable study sites were selected based on the criteria as described in Section 2.3.

### 2.2. DSSAT-CERES-Maize Crop Model

The most recent version of CERES-Maize (DSSAT v4.7) crop model was used to simulate spring maize development, growth and yield in China as a function of weather data, soil characteristics, crop management practices, crop genotype and phenotype (Figure 2). This new version of CERES-Maize (DSSAT v4.7) is the best in comparison to all the other previous versions of CERES-Maize (DSSAT) models since it has more added functions and data that help in the simulation process [19]. According to the report from The United Nations Framework Convention on Climate Change (UNFCCC) [20], this model was cited as one of the best tools that can be integrated with other tools to simulate and evaluate climate change effects on crop production. Input data on crop management practices to be included are: crop type, cultivar, planting date, planting density, planting space and rows, fertilizer type and rate of application, irrigation method and amount and tillage practices [21]. Weather data to be included are: temperature (minimum and maximum), precipitation, humidity and solar radiation. Soil data to be included are: soil type, soil texture, soil pH, soil moisture, soil organic carbon and soil nitrogen levels [22].

### 2.3. Criteria for Site Selection in Validating the Model

To validate the DSSAT-CERES-Maize crop model, we used some measured data to validate the model before carrying out future modeling. The criteria for validating the model were based on the following:Maize cultivars must have been cultivated for a minimum of 3 years and at the same time they should not have been stressed by either diseases, pests, insects or severe climatic events.Records must be available on good field management practices, e.g., adjusting sowing dates, row spacing, cultivar change and irrigation.The location of the study sites should be near the Agricultural Meteorological Experimental Stations (AMESs) to ensure easy accessibility to accurate weather observation data.

### 2.4. Climate Scenario and Climate Data

In this particular study, two climate scenarios RCP4.5 and RCP8.5 were used to investigate the possible effects of anthropogenic and natural activities on future climate variability [23]. Based on a wide range of possible radiative forcing’s (W/m^2^) and greenhouse gas emissions and concentrations, the Fifth Assessment Report (AR5) of the IPCC adopted four RCPs, namely RCP2.6, RCP4.5, RCP6.0 and RCP8.5 [2,24,25]. RCP2.6 is a low-level greenhouse gas emission scenario, RCP4.5 and RCP6.0 are both medium-level greenhouse gas emission scenarios and RCP8.5 is a high-level greenhouse gas emission scenario [23]. The basis of selecting the following two RCPs (RCP4.5 and RCP8.5) for this study was to help compare the impacts of CO_2_ emitted to the atmosphere at different RCP scenarios on climate variables.

### 2.5. Crop Model Input Data

Weather data: To operate the CERES-Maize crop model, weather input data for the selected sites were captured from the China Meteorological Data Network for each of the selected Agricultural Meteorological Experimental Stations (AMESs; http://data.cma.cn/, accessed on 7 May 2022). The weather input data included: daily maximum and minimum air temperatures, daily precipitation, daily humidity and daily solar radiation. The trend in daily solar radiation for each of the selected weather stations was calculated from the observed sunshine hours using the Angstrom Prescott equation [24]. Observed weather data and future climate scenario data were the two types of weather data that were used for calibration and validation of the CERES-Maize crop model. RCP scenario data (RCP4.5 and RCP8.5) generated by GCM-HadGEM2-ES model were adopted for baseline (1986 to 2005) and for the two future periods, the 2030s (2021 to 2040) and the 2050s (2041 to 2060), for maize yield simulations. The datasets on weather and climate were downscaled to a high spatial resolution of 0.5° × 0.5° (55.5 km × 55.5 km) and finally bias-corrected before being incorporated into the CERES-Maize model. Grid weather data (0.5° × 0.5°) were produced by our research group using the PRECISE model authorized by the British Hadley Center, which was downscaled based on the GCM-HadGEM2-ES.

Soil data: The CERES-Maize (DSSAT v4.7) model required soil input data which included soil color, soil texture, soil particle size, soil organic carbon, soil pH, soil nitrogen levels, soil bulk density, soil type, soil cation exchange and soil drainage [26]. These data were acquired from the local AMESs and from the Chinese soil scientific database (http://vdb3.soil.csdb.cn, accessed on 7 May 2022).

Crop observation and management data: The input data on crop management practices for the selected study sites for the period 1985 to 2005 were provided by the Institute of Crop Sciences of Chinese Academy of Agricultural Sciences (http://ics.caas.cn/en/, accessed on 7 May 2022) acquired from the AMESs within the study area. The study station sites that were used to provide the maize data were selected based on the following criteria to ensure credibility of the data output: (1) availability of up-to-date records on maize crop management practices, (2) the maize cultivars to be selected must have been planted for a minimum of 3 years and (3) the maize cultivars should not have been stressed by either diseases, pests, insects or severe climatic events [1]. The crop management input data were used to provide the following parameters for the CERES-Maize (DSSAT v4.7) model: crop type, cultivar, planting date, planting density, planting space, planting rows, fertilizer type, rate of fertilizer application, irrigation method, amount of irrigation, harvesting date and tillage practices [21].

### 2.6. Genetic Coefficients

Genetic coefficients in the DSSAT crop model are outlined as a set of parameters that explain the interrelationship between crop genotype and environmental conditions [23]. There are six genetic coefficients embedded within the CERES-Maize (DSSAT v4.7) model that help characterize a crop cultivar (Table 1, namely P1, P2, P5, G2, G3 and PHINT. These genetic coefficients help to explain the crop phenological stages, growth and development processes involved [2,20]. A coefficient estimator module embedded within DSSAT known as GLUE (Generalized Likelihood Uncertainty Estimation) was developed to help estimate the genotype-specific coefficients of all crop cultivars incorporated within the DSSAT model [1,25].

### 2.7. Model Calibration and Validation

The process of calibration of the CERES-Maize aims to obtain reasonable estimates of model genetic coefficients by comparing simulated data with the observed data [26]. Model validation is a process of assessing the ability of the calibrated model to simulate characteristics of the selected cultivar (Table 2) by comparison between the simulated and observed parameters. If the validation results are unsatisfactory, model calibration would be rerun to obtain a new set of genetic coefficients until the deviations between simulated and observed values are within acceptable limits. In this study, observed data of one representative cultivar from each site were used to calibrate the crop model with GLUE module, and two or more years’ independent data were used to validate the CERES-Maize model. Performance of the model was evaluated by comparing simulated values and observations of days to flowering (from sowing to flowering), days to maturity (length of maize life cycle), maize yields and biomass. The following statistical indicators were used for the evaluation: (i) normalized root mean square error (*NRMSE*), presenting relative error magnitude, and (ii) predicted deviation (*PD*), indicating possible over or underestimation.
$$NRMSE=\frac{{\left[\;\sum_{i=1}^{n}\frac{{\left(S_{i}-O_{i}\right)}^{2}}{n}\;\right]}^{0.5}}{\bar{o}}$$
$$PD_{i}=\left(S_{i}-O_{i\;}\right)/O_{i}$$
where $S_{i}$ and $O_{i}$ are simulated and observed variables, respectively; $\bar{o}$ is the mean value of the observed data; n is the number of comparisons; and i is each comparison. The simulation was considered excellent if NRMSE is <10%, good if NRMSE is >10% and <20%, fair if NRMSE is >20% and <30% and poor if NRMSE is >30% [2]. Additionally, a negative PD value indicates underprediction, while a positive one indicates overprediction [2,21].

#### Method to Manage the Uncertainty of Simulations

The GLUE (Generalized Likelihood Uncertainty Estimation) program is used to estimate genotype-specific coefficients for the DSSAT crop models. It is a Bayesian estimation method that uses Monte Carlo sampling from prior distributions of the coefficients and a Gaussian likelihood function to determine the best coefficients based on the simulated and observed yield values [27]. The uncertainties in predictions of the impact of future climate change on the yield of spring maize in the main production areas of China were evaluated using the existing field corn genotype coefficient and soil parameter database contained within the DSSAT and field data collected from the agrometeorological stations near the study sites [28]. Using the GLUE Select program, more than 3000 runs were initiated so as to refine the results since less than 3000 runs would not likely give reliable results [29]. A sensitivity analysis involving the use of the Morris and Sobol sensitivity analysis methods was performed on the DSSAT v4.7 CERES-Maize model input parameters: cultivar selection, cultivar coefficients, crop management including planting date, spacing and plant density, soil profile and weather data. In general, a sensitivity analysis is the study of how a model evaluates the sensitivity of model outputs based on the variation in the input variables and genotype-specific parameters [30].

The Morris method is a global sensitivity analysis method based on screening analysis, using individually randomized one-factor-at-a-time (OAT) designs, by calculating the input parameters to perform sensitivity analysis on the basic effects of the output results. This method effectively balances the efficiency and accuracy of calculation. It is useful for models with many input parameters and a large computational load [31].

Sobol is a global sensitivity analysis method based on variance, which is determined by decomposing the variance of output variables. It uses the ANOVA-based decomposition to quantitatively evaluate the impact of each input parameter and the interaction between the parameters on the output variable [31]. Additionally, the method of Sobol is able to estimate the total sensitivity index, defined as the sum of all effects (including first-order and higher-order) involving the input factor of interest [32].

## 3. Results

### 3.1. Changes in the Major Meteorological Elements by the 2030s and 2050s in Northeast China

Table 3 shows projected changes in annual average temperature, precipitation and solar radiation relative to the baseline under RCP4.5 and RCP8.5 at Boli, Hailun, Tonghua, Huadian, Wafangdian and Dengta during the future periods (2030s and 2050s) compared with the baseline (1986–2005). In Heilongjiang, Jilin and Liaoning provinces, where the six study sites are located, the projected mean temperature during the 2030s and 2050s was set to change by a range of between −2.14 and 2.56 °C and −1.11 and 3.64 °C, respectively, under RCP4.5. The change in mean temperature during the 2030s and 2050s under RCP8.5 for the study sites ranged between −2.21 and 2.39 °C and −0.85 and 3.85 °C, respectively, which was higher than that under RCP4.5 especially for the 2050s due to the high CO_2_ concentration estimated for this period. Under RCP4.5, the annual average precipitation was projected to increase by a range of between 6.6 and 24.68% and −1.94 and 25.24% during the 2030s and 2050s, respectively. While precipitation under RCP8.5 was projected to change by a range of between −6.06 and 23.01% and −3.35 and 23.43% for the future periods 2030s and 2050s, respectively, which is an increase compared to RCP4.5 for the future periods in all the six study sites. The solar radiation compared to the baseline for the six study sites was projected to decrease by a range of between −6.0 and −23.24% and −4.82 and −23.44% for the 2030s and 2050s durations, respectively, under RCP4.5 and projected to further decrease with a range of between −7.32 and −24.58% and −7.35 and −24.43% for the 2030s and 2050s durations, respectively, under RCP8.5.

### 3.2. Model Calibration and Validation

#### Genetic Parameters Estimation

Calibration results of the six maize cultivars in Hailun, Boli, Huadian, Tonghua, Wafangdian and Dengta using the CERES-Maize model (Table 1) projected a set of six genetic cultivar coefficients (Table 4) for each of the maize cultivars. Six genetic coefficients (P1, P2, P5, G2, G3 and PHINT) are used by the CERES-Maize model to characterize or define the maize cultivar. Table 1 provides the definitions of the six genetic parameters, with four of the parameters (P1, P2, P5 and PHINT) controlling the timing of the phenological stages, while the other two parameters (G2 and G3) characterizing the potential yield under optimal conditions. Two statistical indicators were used for the evaluation: (i) normalized root mean square error (NRMSE), presenting relative error magnitude, and (ii) predicted deviation (PD), indicating possible over- or underestimation. The statistical indicators were calculated as shown in Section 2.3. The simulated values for maize yield were considered excellent if NRMSE is <10%, good if NRMSE is >10% and < 20%, fair if NRMSE is >20% and <30% and poor if NRMSE is >30% (Table 4). The NMRSE for the time to flowering, time to maturity and maize yield for all the six maize cultivars ranged between 0.5 and 9.9% which is considered excellent apart from only one site in Wafangdian which had a value of 10.8% which is considered good (Table 4). The predicted deviation values for all the six maize cultivars regarding the difference between simulated and observed time to flowering, time to maturity and maize yield ranged between −5.1 and 8.8% (Table 4). In conclusion, the CERES-Maize model was able to simulate grain yield under various agricultural management strategies and soil conditions, which suggests that this model could be applied in NEC.

### 3.3. Impact of Future Climate Change on Maize Yield by the 2030s and 2050s

#### 3.3.1. Impact on the Time-to-Flowering

The time-to-flowering which is the period from sowing to flowering in days was compared based on the six different genetic coefficient parameters to establish the variation based on different climate change scenarios. The modeling results showed that maize phenology for the six cultivars based on the different genetic coefficients under scenarios RCP4.5 and RCP8.5 presented a decreasing trend for the number of days to flowering (Figure 3) if no adaptation measure was adopted. This reduction in the number of days to flowering is attributed to the projected increase in temperature and reduction in solar radiation under RCP4.5 and 8.5 for the future period between 2021 to 2060 (Table 3). Additionally, the difference in CO_2_ levels between RCP4.5 (medium-level greenhouse gas emission) compared to RCP8.5 (high-level greenhouse gas emission) contributed to the large reduction in the number of days to flowering for maize under RCP8.5 versus RCP4.5. The reduction in the number of days to flowering ranged from 1 day under parameter 6 in Hailun for the 2030s period to 11 days under parameter 3 in Boli for the 2050s period under RCP4.5, and also under RCP8.5 it ranged from an increase of 1 day under parameter 4 in Tonghua for the 2030s period to a reduction of 14 days under parameter 2 in Huadian for the 2050s period (Figure 3). Both Boli and Huadian recorded the largest reduction in the number of days to flowering in Heilongjiang and Jilin provinces of N.E China, respectively, and this was mainly attributed to the projected rise in mean temperature based on both scenarios RCP4.5 and RCP8.5 (Table 3). In Liaoning Province, Wafangdian recorded 12 days, the largest reduction in the number of days to flowering compared to Dengta which had 8 days and this was mainly attributed to the projected increase in temperature under RCP4.5 and 8.5 for the future period between 2021 to 2060 (Table 3).

#### 3.3.2. Impact on the Time-to-Maturity

The time-to-maturity, which is the length of the maize life cycle in days, was compared based on the six different genetic coefficient parameters to establish the variation based on different climate change scenarios. The simulated days to maturity for all the study sites based on the six different genetic coefficient paraments for each site recorded a decrease that ranged from 7 days under parameter 6 in Hailun in the 2030s to 36 days under parameter 1 and 3 in Hailun in the 2050s both under RCP4.5, also under RCP8.5 the simulated days to maturity projected a decline ranging from 11 days under parameter 6 in Hailun in the 2030s to 38 days under parameter 1 and 2 in Hailun and Wafangdian, respectively, in the 2050s (Figure 4). As a result of the warming climate projected for all the study sites in Northeast China in the 2030s and 2050s, both based on RCP scenarios of 4.5 and 8.5 (Table 3), crop development has been hastened and consequently inducing earlier flowering and maturity dates (Figure 3 and Figure 4) under the different parameters. The other contributing factor was the variation in the level of CO_2_ between RCP4.5 (medium-level greenhouse gas emission) and RCP8.5 (high-level greenhouse gas emission) that contributed to the highest reduction in the time-to-maturity for maize under RCP8.5 versus RCP4.5. In Heilongjiang Province, the highest predicted reduction in the time-to-maturity was in Hailun under parameter 1 at 38 days compared to Boli which was predicted to decrease 35 days under parameter 1 for the future period between 2021 to 2060. In Jilin Province, the highest predicted reduction in the time-to-maturity was in Huadian under parameter 2 at 33 days compared to Tonghua in which it was 32 days under parameter 4 for the future period between 2021 to 2060. In Liaoning Province, the highest predicted reduction in the time-to-maturity was at 38 days in Wafangdian under parameter 2 compared to Dengta where the predicted reduction was 35 days under parameter 2 for the future period between 2021 to 2060.

#### 3.3.3. Impact on Maize Yield

The percentage change in maize yield relative to the baseline was calculated based on the average maize yields for the 2030s and 2050s in reference to the future climate change scenarios RCP4.5 and RCP8.5 without considering any adaptation measure for the different genetic coefficients for all the six study sites (Figure 5). The simulation results showed that the overall mean maize yield would have a percentage decrease that ranged from 2.5% in Dengta under parameter 1 for the 2030s period to 25.4% in Wafangdian under parameter 3 for the 2050s period under RCP4.5. Moreover, under RCP8.5, the predicted percentage reduction in mean yield would be even higher, ranging from 2.9% in Dengta under parameter 4 for the 2030s period to 26.4% in Boli under parameter 6 for the 2050s (Figure 5). In Heilongjiang Province, the highest predicted reduction in maize yield was in Boli under parameter 6 at 26.4% compared to Hailun in which yield reduction was predicted to be 17.5% under parameter 3 for the future period between 2021 and 2060. In Jilin Province, the highest reduction in predicted maize yield was in Huadian under parameter 1 at 26.1% compared to Tonghua with 20.7% reduction under parameter 1 for the future period between 2021 and 2060. In Liaoning Province, the highest predicted reduction in maize yield level was in Wafangdian under parameter 3 at 26.3% compared to Dengta with a 10.5% reduction under parameter 6 for the future period between 2021 and 2060. Because maize is a C_4_ plant, the impact of elevated CO_2_ under RCP8.5 (high-level greenhouse gas emission) compared to RCP4.5 (medium-level greenhouse gas emission) did not have any significant impact on the maize yield since the highest reductions in maize yield levels were recorded under RCP8.5. The other factor that contributed to the high reduction in maize yield under RCP8.5 for the future periods of the 2030s and 2050s was attributed to the projected increase and reduction in temperature and solar radiation, respectively, under RCP8.5 compared to RCP4.5 for the future period between 2021 and 2060 (Table 3).

### 3.4. Effect of Adaptation Measures on Maize Yield Based on Different Model Parameters

#### 3.4.1. Effect of Changing Planting Dates on Maize Yield

As shown in Figure 6, maize yield at Boli, Hailun, Tonghua, Huadian, Wafangdian and Dengta study sites would decrease by 0 to 10.1% if the planting dates were advanced by 5 to 15 days under RCP4.5 for the 2030s period, while maize yield would increase by 0 to 12.3% if the planting dates were delayed by 5 to 15 days under RCP4.5 for the 2030s period based on all the different genetic coefficient parameters. According to the simulation, delaying plating dates by 5 to 15 days would result in grain yield increases since this would help the maize crop avoid thermal stress during its key developmental stages. The level of grain yield increases also varied based on the two RCP scenarios, i.e., RCP4.5 (medium-level greenhouse gas emission) and RCP8.5 (high-level greenhouse gas emission) that had varying CO_2_ levels. The highest reduction and increase in maize yield after advancing and delaying the planting dates by 5 to 15 days were recorded in Jilin followed by Liaoning and then Heilongjiang Province.

According to Figure 7, maize yield at Boli, Hailun, Tonghua, Huadian, Wafangdian and Dengta study sites would decrease by 0 to 14.8% if the planting dates were advanced by 5 to 15 days under RCP4.5 for the 2050s period, while maize yield would increase by 0 to 19.4% if the planting dates were delayed by 5 to 15 days under RCP4.5 for the 2050s period based on all the different genetic coefficient parameters. The highest reduction in maize yield after advancing the planting dates by 5 to 15 days was recorded in Liaoning followed by Heilongjiang and then Jilin Province. While the highest increase in maize yield after delaying the planting dates by 5 to 15 days was recorded in Jilin followed by Heilongjiang and then Liaoning Province.

As shown in Figure 8, maize yield at Boli, Hailun, Tonghua, Huadian, Wafangdian and Dengta would decrease by 0 to 9.5% if the planting dates were advanced by 5 to 15 days under RCP8.5 for the 2030s period, while maize yield would increase by 0 to 11.9% if the planting dates were delayed by 5 to 15 days under RCP8.5 for the 2030s period based on all the different genetic coefficient parameters. The highest reduction in maize yield after advancing the planting dates by 5 to 15 days was recorded in Jilin followed by Heilongjiang and then Liaoning Province. While the highest increase in maize yield after delaying the planting dates by 5 to 15 days was recorded in Jilin followed by Liaoning and then Heilongjiang Province.

According to Figure 9, maize yield at Boli, Hailun, Tonghua, Huadian, Wafangdian and Dengta study sites would decrease by 0 to 14% if the planting dates were advanced by 5 to 15 days under RCP8.5 for the 2050s period, meanwhile if the planting dates were delayed by 5 to 15 days under RCP8.5 for the 2050s period, maize yield would increase by 0 to 21.6% based on all the different genetic coefficient parameters. The highest reduction in maize yield after advancing the planting dates by 5 to 15 days was recorded in Heilongjiang followed by Liaoning and then Jilin Province. While the highest increase in maize yield after delaying the planting dates by 5 to 15 days was recorded in Heilongjiang followed by Jilin and then Liaoning Province.

#### 3.4.2. Effect of Adding Irrigation Practices on Maize Yield

Under irrigation practices and early planting by 5 to 15 days, the maize yield at Boli, Hailun, Tonghua, Huadian, Wafangdian and Dengta sites declined by 0 to 8.5% under RCP4.5 for the 2030s period, meanwhile under irrigation practices and delayed planting by 5 to 15 days at the Boli, Hailun, Tonghua, Huadian, Wafangdian and Dengta sites, the maize yield levels increased by 0 to 12.2% under RCP4.5 for the 2030s period (Figure 10). The highest reduction in maize yield after advancing the planting dates by 5 to 15 days was recorded in Jilin followed by Heilongjiang and then Liaoning Province. While the highest increase in maize yield after delaying the planting dates by 5 to 15 days was recorded in Jilin followed by Heilongjiang and then Liaoning Province.

The maize yield levels for Boli, Hailun, Tonghua, Huadian, Wafangdian and Dengta under irrigation practices under RCP4.5 for the 2050s period decreased by 0 to 11% when the planting dates were advanced by 5 to 15 days, but when delaying the planting dates by 5 to 15 days the maize yields for Boli, Hailun, Tonghua, Huadian, Wafangdian and Dengta increased by 0 to 24.4% under RCP4.5 for the 2050s period (Figure 11). The highest reduction in maize yield after advancing the planting dates by 5 to 15 days was recorded in Liaoning followed by Jilin and then Heilongjiang Province. While the highest increase in maize yield after delaying the planting dates by 5 to 15 days was recorded in Heilongjiang followed by Jilin and then Liaoning Province.

The maize yield levels for Boli, Hailun, Tonghua, Huadian, Wafangdian and Dengta under RCP8.5 for the future 2030s period under irrigation practices based on advanced planting dates of 5 to 15 days showed a decrease of between 0 and 13.3%, meanwhile after adjusting to the delayed planting dates of 5 to 15 days, the maize yield levels for the Boli, Hailun, Tonghua, Huadian, Wafangdian and Dengta sites increased by 0 to 13.8% (Figure 12). The highest reduction in maize yield after advancing the planting dates by 5 to 15 days was recorded in Liaoning followed by Jilin and then Heilongjiang Province. While the highest increase in maize yield after delaying the planting dates by 5 to 15 days was recorded in Jilin followed by Liaoning and then Heilongjiang Province.

Under RCP8.5 for the future 2050s period under irrigation practices based on early planting of 5 to 15 days for Boli, Hailun, Tonghua, Huadian, Wafangdian and Dengta, the maize yield levels decreased by 0 to 10.7%, meanwhile delaying the planting dates by 5 to 15 days for Boli, Hailun, Tonghua, Huadian, Wafangdian and Dengta, the maize yield levels increased by 0 to 19% (Figure 13). The highest reduction in maize yield after advancing the planting dates by 5 to 15 days was recorded in Jilin followed by Heilongjiang and then Liaoning Province. While the highest increase in maize yield after delaying the planting dates by 5 to 15 days was recorded in Jilin followed by Liaoning and then Heilongjiang Province.

#### 3.4.3. Effect of Replacing Cultivar on Maize Yield

According to the impact simulations, the late-maturing cultivar Dongdan60 at the Dengta site had a longer growing period of 141 days compared to the shorter season cultivars 4zao6 at Boli and Limin15 at Huadian that had 136 days and 139 days, respectively, and a lower decrease in yield relative to the cultivars at higher latitudes. Thus, this cultivar was selected to be an adaptation measure to replace the shorter season cultivars 4zao6 at Boli and Limin15 at Huadian, which were predicted to suffer higher yield reductions under future climate conditions. A late-maturing cultivar Dongdan60 at the Dengta site in Liaoning Province of Northeast China was selected to replace the early and medium-maturing maize cultivars 4zao6 and Limin15 grown at Boli in Heilongjiang Province and Huadian in Jilin Province, respectively (Figure 14). According to the simulation results, if the local maize cultivar 4zao6 at Boli is replaced by Dongdan60, the maize yield would increase by 18.9% under RCP4.5 for the 2030s period, increase by 23.6% under RCP4.5 for the 2050s period, increase by 16.5% under RCP8.5 for the 2030s period and increase by 15.6% under RCP8.5 for the 2050s period based on parameter 4 (optimal treatment parameter) (Figure 14). In addition, changing the maize cultivar Limin15 at Huadian site with Dongdan60 would increase the maize yields by 13.9% under RCP4.5 for the 2030s period, increase by 15.9% under RCP4.5 for the 2050s period, increase by 10.8% under RCP8.5 for the 2030s period and increase by 9.7% under RCP8.5 for the 2050s period based on parameter 1 (optimal treatment parameter) (Figure 14). Compared to the local cultivars, the new late-maturing cultivar introduced from the lower altitude site under the same climate and soil conditions caused the maize yields to increase significantly. The late-maturing cultivar cultivated at a lower altitude has a longer growing period and as such it would take advantage of the increased temperatures resulting in yield increase as opposed to the local maize cultivar which would be affected by the increased temperatures resulting in yield reduction. Thus, more benefits would result from the development of new crop cultivars that are tolerant of high temperatures with higher thermal requirements.

#### 3.4.4. Effect of Adopting Multiple Measures on Maize Yield

Upon adopting early planting dates, irrigation practices and cultivar change, which entailed adopting the late-maturing cultivar Dongdan60 to replace the early and medium-maturing maize cultivars 4zao6 at Boli and Limin15 grown at Boli in Heilongjiang Province and Huadian in Jilin Province, respectively (Figure 15), an exponential percentage increase in maize yield was recorded at both sites. In Boli, upon adoption of all three adaptation measures, the maize yield levels were projected to increase by 38.9% under RCP4.5 for the 2030s period, increase by 43.6% under RCP4.5 for the 2050s period, increase by 36.5% under RCP8.5 for the 2030s period and increase by 35.6% under RCP8.5 for the 2050s period based on parameter 4 (optimal treatment parameter) (Figure 15). In Huadian, upon adoption of all three adaptation measures, the maize yield levels were projected to increase by 27.9% under RCP4.5 for the 2030s period, increase by 29.9% under RCP4.5 for the 2050s period, increase by 24.8% under RCP8.5 for the 2030s period and increase by 23.7% under RCP8.5 for the 2050s period based on parameter 1 (optimal treatment parameter) (Figure 15). As a result, the benefits of climate change such as the increased temperatures can be exploited by adopting maize cultivars with a longer growth period and integrating the other two adaptation measures of changing planting dates and irrigation practices.

## 4. Discussion

### 4.1. Possible Future Meteorological Elements in NEC

At the six representative sites in NEC, analyses indicate that the projected annual average temperature during the 2030s and 2050s under RCP4.5 and RCP8.5 is predicted to change by a range of −2.21 to 3.85 °C. This is consistent with the findings of Yin et al., (2015) [22] in which an increase of 2 °C is predicted to occur in over 60% of all regions in China by the end of the 21st century. Previous studies have predicted that solar radiation in NEC will decrease [7]. In agreement with this, based on our projected simulations for the 2030s and 2050s periods under RCP4.5 and RCP8.5 NEC, a change of −7.35 to 24.58% was predicted. The annual average precipitation according to our simulation was projected to change by −6.06 to 25.24% for the 2030s and 2050s periods under RCP4.5 and RCP8.5, which agrees with previous findings for NEC [1] of projected increases in precipitation by 6.8, 11.1 and 14.4% and 7.2, 12.9 and 23.7% under RCP2.6 and RCP8.5 for the 2020s, 2050s and 2080s periods, respectively.

### 4.2. Impact of Future Climate Change on Maize Yield under Different Parameters

In NEC, maize production is largely dependent on rainfall and according to Na et al., (2021) [11] the impact of changing climate will have a significant effect on maize yield. The agriculture systems in Northeast China are more sensitive to climate warming and this has limited the adaptability of agro-ecosystems to the changing climate [33]. As a result of climate warming in NEC, the resultant effect has been a recorded variation in maize production due to accelerated crop development leading to a reduction in the length of the growing season [2]. According to our results, increased temperatures and a decrease in sunshine hours and precipitation levels at Boli, Hailun, Tonghua, Huadian, Wafangdian and Dengta for the 2030s and 2050s periods based on RCP4.5 and RCP8.5 climate scenarios would shorten the maize growth durations by (1–38 days) and this would result in a reduction in maize yield by (2.5–26.4%). Our results were consistent with the findings of Li et al. [34] where maize yield reduction was recorded in some regions of China under projected global warming by 1.5 °C and 2.0 °C.

### 4.3. Effect of Adaptaion Measures on Maize Yield

Adaptation measures to mitigate the negative effect of climate change are very critical in ensuring food security [11,12]. However, climate warming will increase temperatures and extend the length of the potential growing period of crops if other resources are not limited. Our findings indicated that an effective management option to counteract the negative warming effects and reduce maize yield loss due to the sensitive nature of cultivars to high temperatures is to use early planting, change to more heat-tolerant cultivars and add irrigation practice [35]. Upon adoption of the adaptation measures tested in this study, the maize yield levels in NEC were projected to increase by a range of 23.7 to 43.6% for the 2030s and 2050s period under RCP4.5 and RCP8.5. Our findings were consistent with the studies of Xu et al. [16] and Na et al. [11] which indicated that the use of maize varieties with longer growing periods among other adaptation measures can help increase maize yield under future climate change. Future research studies should analyze the influence of soil conditions, human resource, market price and local production levels that might influence the variation in maize yields, factors that were not considered in this study.

## 5. Conclusions

Using the CERES-Maize model v4.7, six sets of genetic parameters were generated and the results showed that time-to-flowering and time-to-maturity of maize would be shortened by 1 to 14 days and 7 to 38 days, respectively, from 2021 to 2060 relative to 1986 to 2005, based on the future climate at six representative sites in NRC. Additionally, the simulation results showed that increased temperatures and a decrease in precipitation levels and sunshine hours during the 2030s and 2050s would shorten maize growth durations and this would result in more yield reduction under RCP 8.5 than RCP 4.5. Simulation indicated that adaptation measures, including changing planting dates, adding irrigation practice and replacing cultivar, would increase maize yield to varying degrees of between (23.7 and 43.6%) at both Boli and Huadian sites in Northeast China for the 2030s and 2050s periods under RCP 4.5 and RCP 8.5.

## Figures and Tables

**Figure 1 plants-11-01634-f001:**
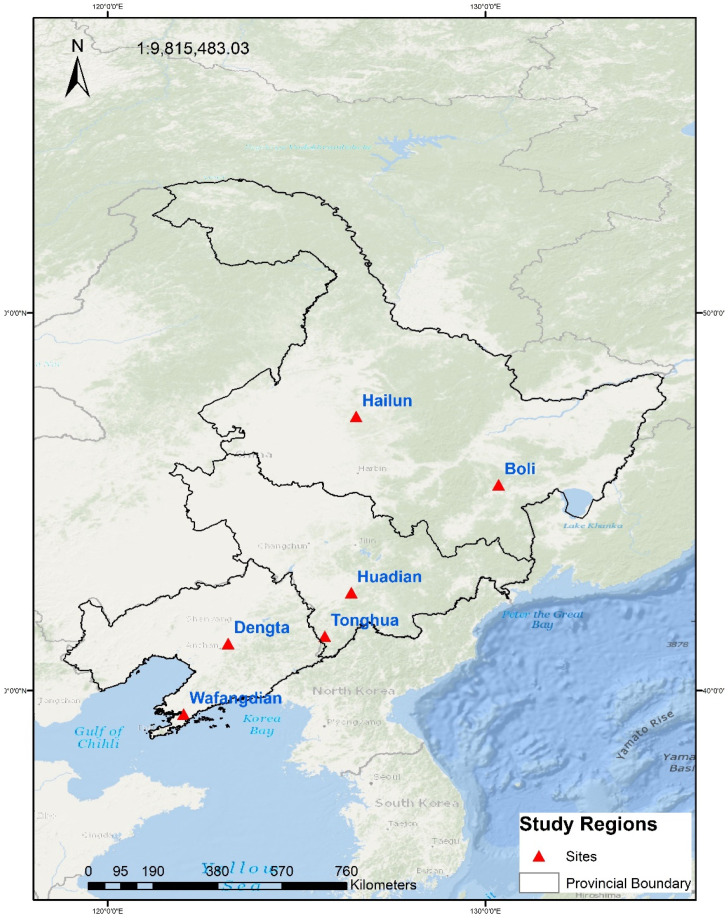
Geographical distribution of study sites in Northeast China.

**Figure 2 plants-11-01634-f002:**
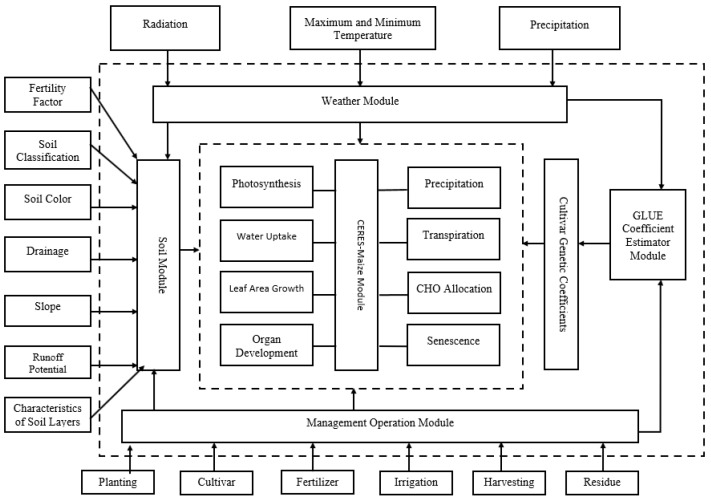
Sketch of the structure of CERES crop model.

**Figure 3 plants-11-01634-f003:**
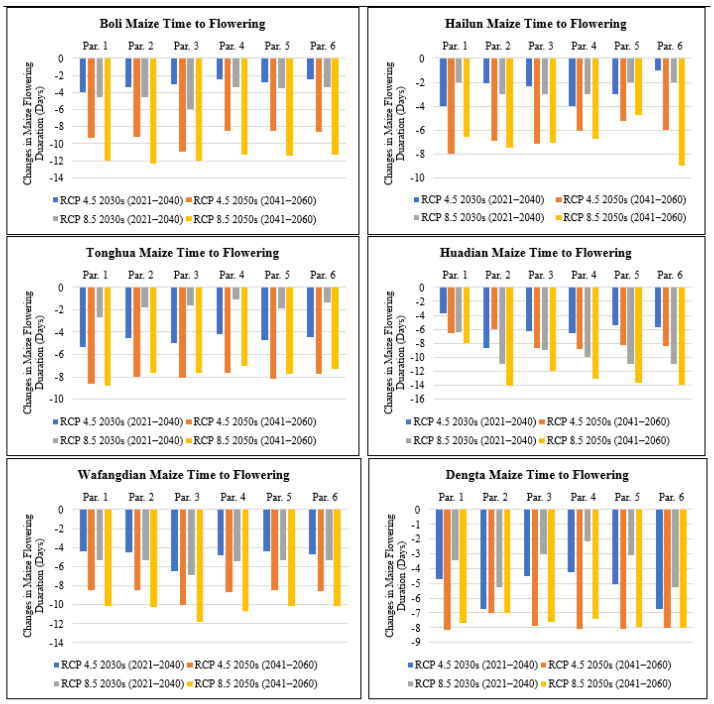
Changes in maize time-to-flowering (days) under future climate scenarios based on different treatment parameters compared with the baseline at Boli, Hailun, Tonghua, Huadian, Wafangdian and Dengta.

**Figure 4 plants-11-01634-f004:**
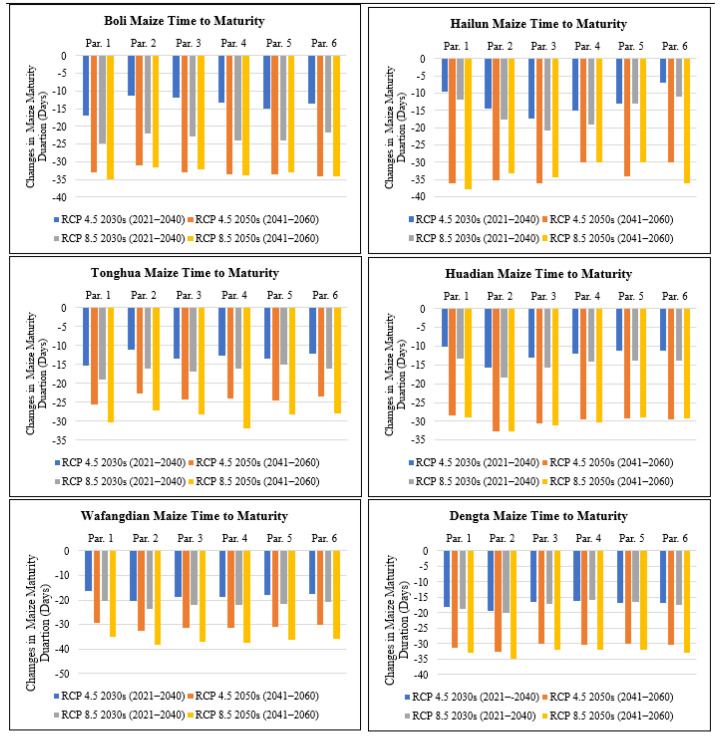
Changes in maize time-to-maturity (days) under future climate scenarios based on different treatment parameters compared with the baseline at Boli, Hailun, Tonghua, Huadian, Wafangdian and Dengta.

**Figure 5 plants-11-01634-f005:**
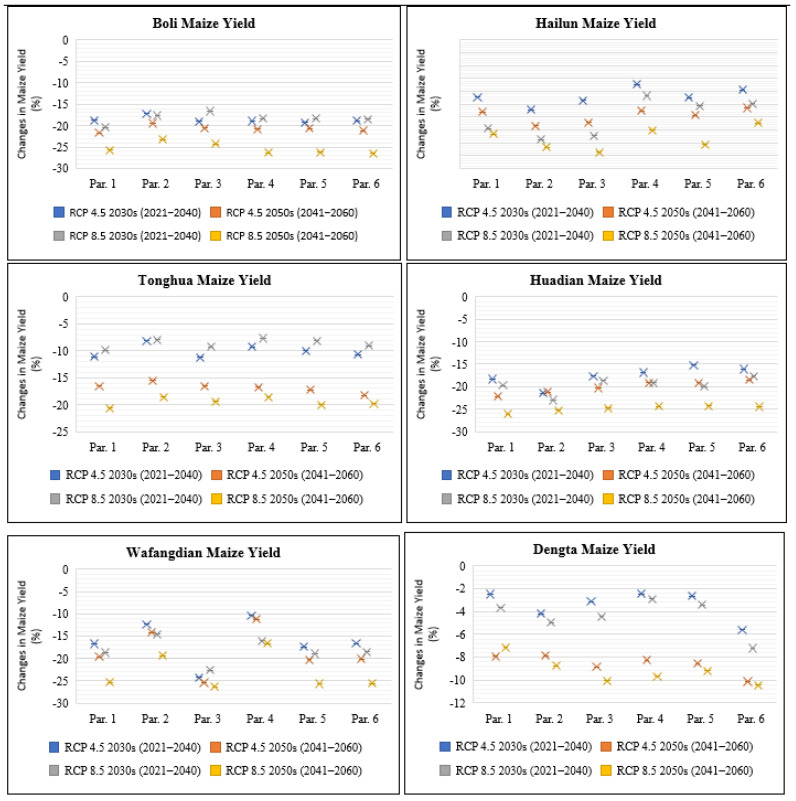
Changes in maize yield (%) under future climate scenarios based on different treatment parameters compared with the baseline at Boli, Hailun, Tonghua, Huadian, Wafangdian and Dengta.

**Figure 6 plants-11-01634-f006:**
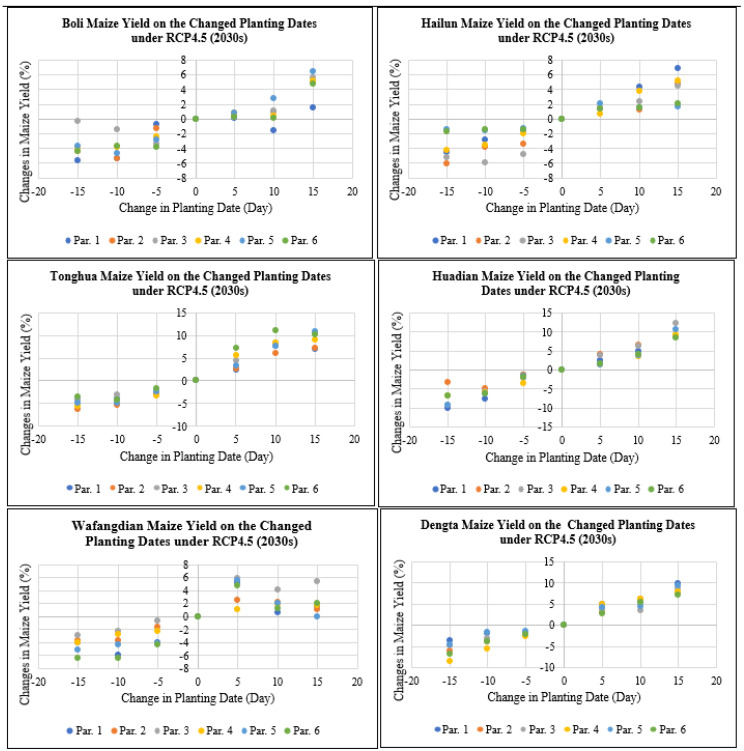
Maize yield on the changed planting dates under RCP4.5 (2030s) based on different treatment parameters compared with the baseline at Boli, Hailun, Tonghua, Huadian, Wafangdian and Dengta.

**Figure 7 plants-11-01634-f007:**
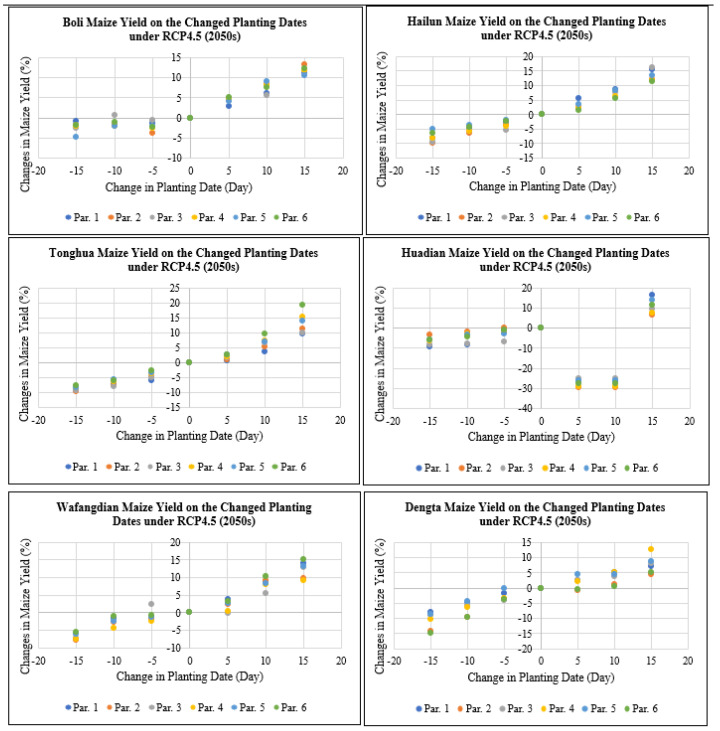
Maize yield on the changed planting dates under RCP4.5 (2050s) based on different treatment parameters compared with the baseline at Boli, Hailun, Tonghua, Huadian, Wafangdian and Dengta.

**Figure 8 plants-11-01634-f008:**
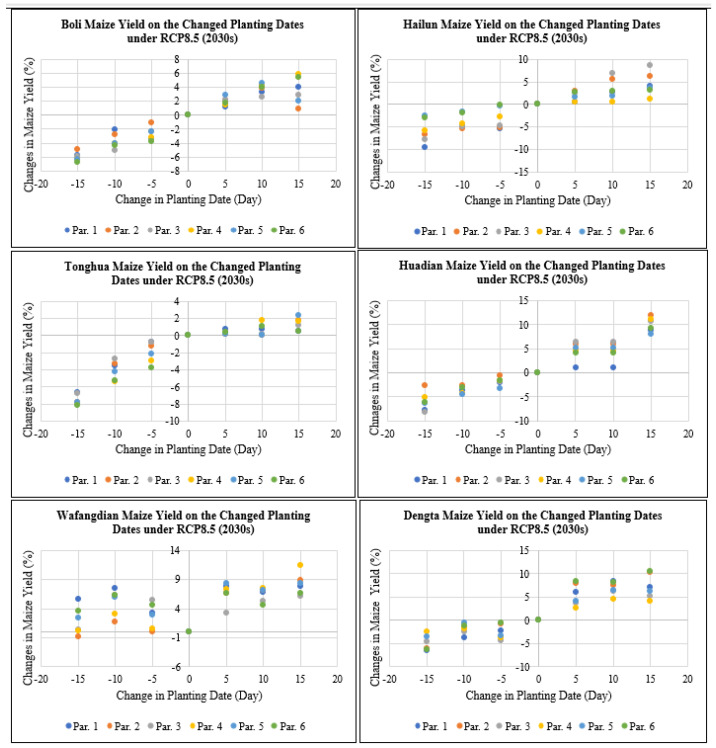
Maize yield on the changed planting dates under RCP8.5 (2030s) based on different treatment parameters compared with the baseline at Boli, Hailun, Tonghua, Huadian, Wafangdian and Dengta.

**Figure 9 plants-11-01634-f009:**
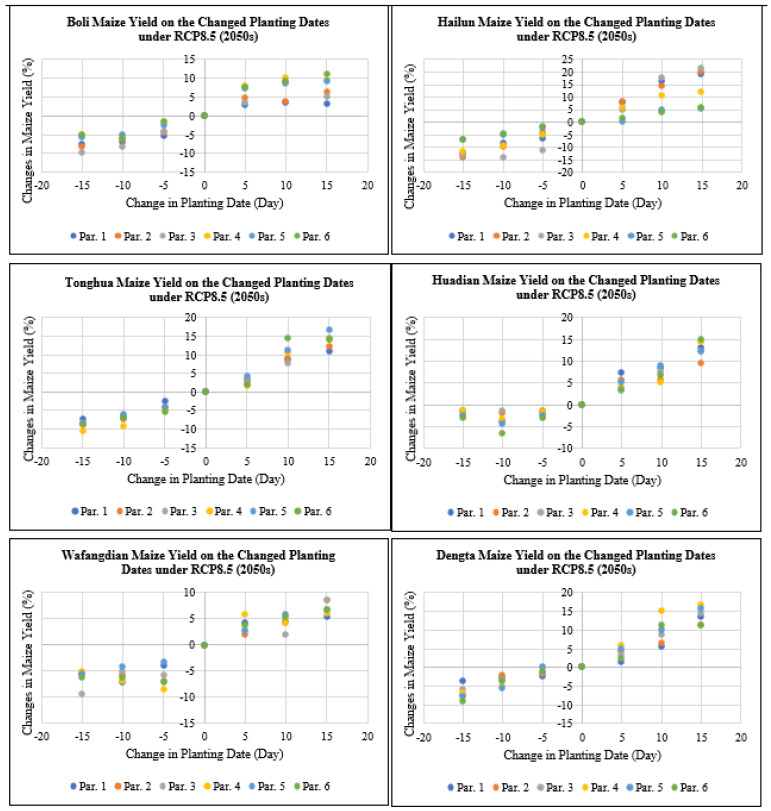
Maize yield on the changed planting dates under RCP8.5 (2050s) based on different treatment parameters compared with the baseline at Boli, Hailun, Tonghua, Huadian, Wafangdian and Dengta.

**Figure 10 plants-11-01634-f010:**
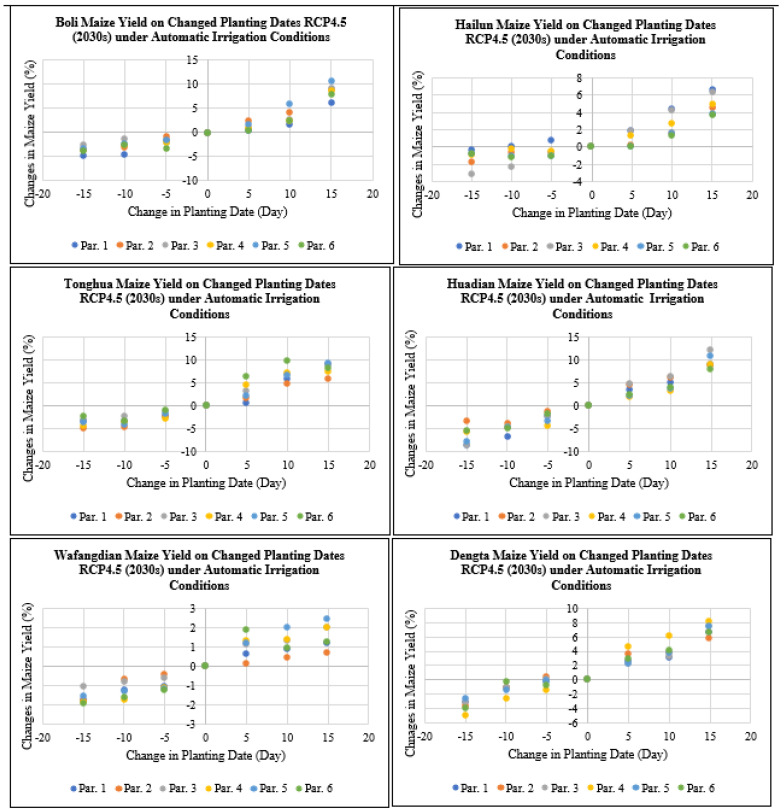
Maize yield on the changed planting dates under RCP4.5 (2030s) under automatic irrigation based on different treatment parameters compared with the baseline at Boli, Hailun, Tonghua, Huadian, Wafangdian and Dengta.

**Figure 11 plants-11-01634-f011:**
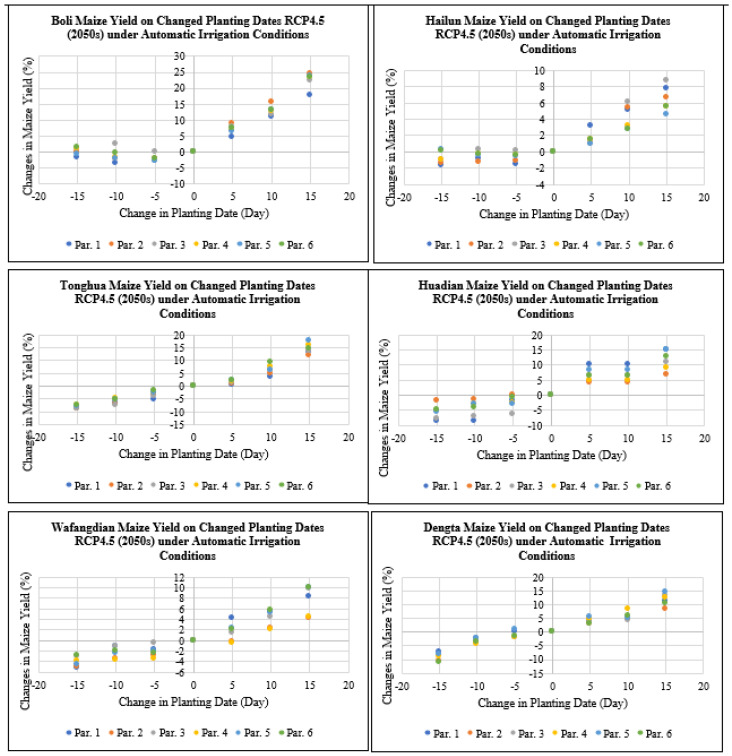
Maize yield on the changed planting dates under RCP4.5 (2050s) under automatic irrigation based on different treatment parameters compared with the baseline at Boli, Hailun, Tonghua, Huadian, Wafangdian and Dengta.

**Figure 12 plants-11-01634-f012:**
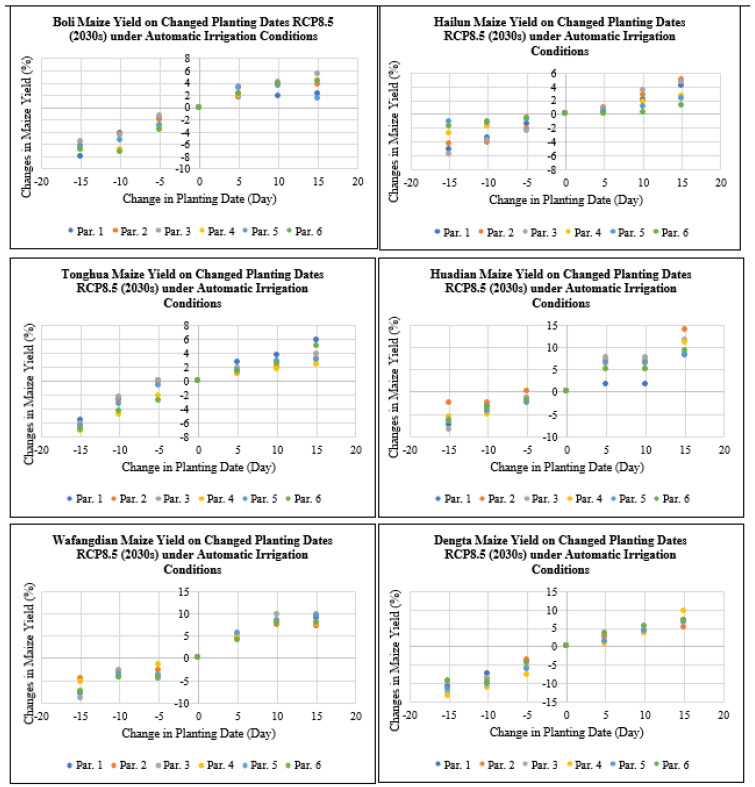
Maize yield on the changed planting dates under RCP8.5 (2030s) under automatic irrigation based on different treatment parameters compared with the baseline at Boli, Hailun, Tonghua, Huadian, Wafangdian and Dengta.

**Figure 13 plants-11-01634-f013:**
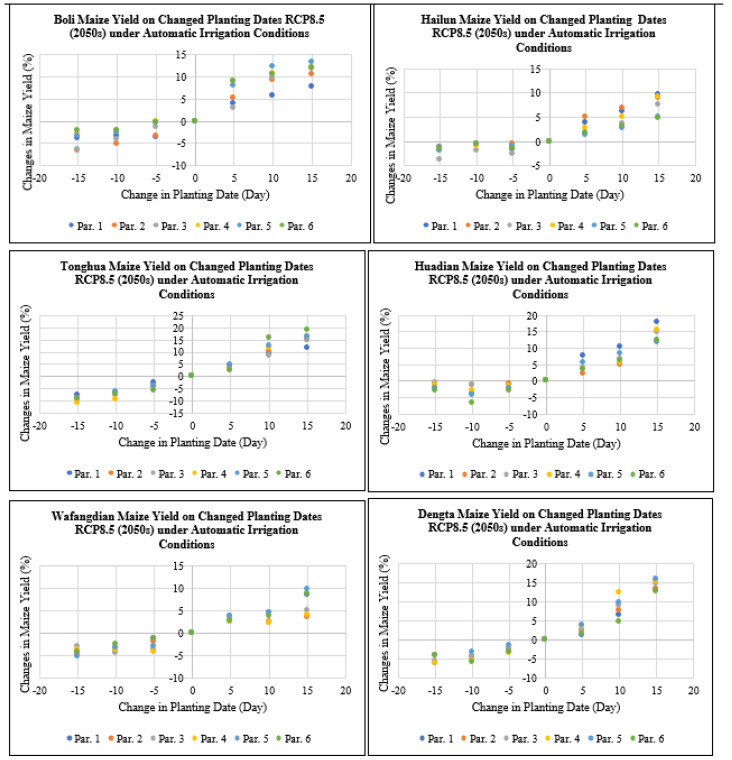
Maize yield on the changed planting dates under RCP8.5 (2050s) under automatic irrigation based on different treatment parameters compared with the baseline at Boli, Hailun, Tonghua, Huadian, Wafangdian and Dengta.

**Figure 14 plants-11-01634-f014:**
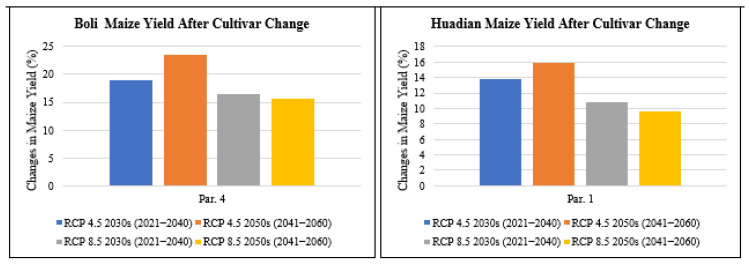
Changes in maize yield (%) after cultivar change with Dongdan60 under future climate scenarios based on one set of optimal treatment parameters compared with the baseline for Boli and Huadian.

**Figure 15 plants-11-01634-f015:**
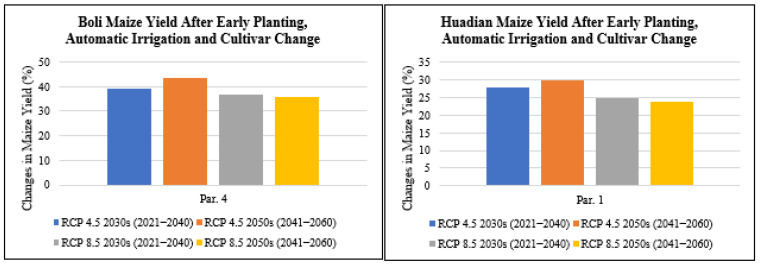
Changes in maize yield (%) after early planting, automatic irrigation and cultivar change with Dongdan60 under future climate scenarios based on one set of optimal treatment parameters compared with the baseline for Boli and Huadian.

**Table 1 plants-11-01634-t001:** Definition of genetic coefficients used in the CERES-Maize (DSSAT v4.7) model.

Coefficient	Definition	Unit
P1	Thermal time from seedling emergence to the end of the juvenile phase (above a base temperature of 8°C) during which the plant is not responsive to changes in photoperiod	°C days
P2	Extent to which development is delayed for each hour increase in photoperiod above the longest photoperiod at which development proceeds at a maximum rate (which is considered to be 12.5 h).	days
P5	Thermal time from silking to physiological maturity (expressed in degree days above a base temperature of 8°C).	°C days
G2	Maximum possible number of kernels per plant.	Number
G3	Kernel filling rate during the linear grain filling stage and under optimum conditions.	Mg day^−1^
PHINT	Phyllochron interval: the interval in thermal time between successive leaf tip appearances	°C days

**Table 2 plants-11-01634-t002:** Selected maize cultivar study sites for calibration and validation within the DSSAT v4.7 crop model.

Province	Site	Latitude (N)	Longitude (E)	Altitude (m)	Maize Cultivar	Type of Maturity	Years
Heilongjiang	Boli	45.45	130.35	220.5	4zao6	Early	2005, 2006 *, 2008
	Hailun	47.26	126.58	239.2	Haiyu 6	Medium	2005, 2006, 2007 *
Jilin	Tonghua	41.4	125.44	384.3	Jidan 159	Medium	2006, 2008 *, 2009
	Huadian	42.59	126.45	263.3	Limin15	Medium	2006, 2007, 2008 *
Liaoning	Wafangdian	39.38	122.01	118.5	Lianyu 6	Late	2006 *, 2007, 2008
	Dengta	41.25	123.19	42	Dongdan 60	Late	2006 *, 2007, 2008

* Indicates the years (observed records) used for calibration of the crop model.

**Table 3 plants-11-01634-t003:** Study sites projected changes in annual average temperature, precipitation and solar radiation relative to the baseline under RCP4.5 and RCP8.5.

Province	Study Site	Mean Temperature (°C)			Precipitation (mm) (%)		Solar Radiation (MJ m^−2^ d^−1^) (%)	
Heilongjiang	Boli	**Scenarios**	**2030s**	**2050s**	**2030s**	**2050s**	**2030s**	**2050s**
		RCP 4.5	0.18	1.30	24.68	21.20	−8.72	−8.54
		RCP 8.5	0.08	1.57	15.01	13.51	−8.72	−10.26
	Hailun	RCP 4.5	1.70	2.92	12.98	−1.94	−6.00	−4.82
		RCP 8.5	1.64	3.19	5.52	4.15	−7.32	−7.35
Jilin	Tonghua	RCP 4.5	−2.14	−1.11	23.16	25.24	−23.24	−23.44
		RCP 8.5	−2.21	−0.85	12.55	22.73	−24.58	−24.43
	Huadian	RCP 4.5	2.56	3.64	6.60	2.11	−9.89	−10.05
		RCP 8.5	2.39	3.85	−6.06	−0.07	−11.78	−11.28
Liaoning	Wafangdian	RCP 4.5	1.78	2.87	8.65	15.76	−16.15	−15.51
		RCP 8.5	1.98	3.37	23.01	23.43	−16.97	−16.22
	Dengta	RCP 4.5	1.94	3.03	8.33	2.58	−9.15	−9.51
		RCP 8.5	1.92	3.35	−4.28	−3.35	−10.50	−9.84

**Table 4 plants-11-01634-t004:** Calculated genetic coefficients for the study sites and respective maize cultivars.

Calculated Genetic Coefficients of Each Maize Cultivar									Time-to-Flowering		Time-to-Maturity		Maize Yield	
Site	Parameter	Cultivar *	P1	P2	P5	G2	G3	PHINT	NRMSE (%)	PD (%)	NRMSE (%)	PD (%)	NRMSE (%)	PD (%)
Boli	1	4zao6	195	0.13	606	979	7.63	49	4.6	1.4	3.2	3.7	4.5	1.5
Boli	2	4zao6	172	0.50	655	978	7.09	49	2	−1.4	2.8	1.5	1.6	−4.2
Boli	3	4zao6	155	0.52	682	983	7.06	49	2.7	1.4	2.8	3.7	1	−3.0
Boli	4	4zao6	165	0.52	648	980	7.20	49	2	−1.4	3.2	1.5	1.5	−1.3
Boli	5	4zao6	155	0.80	650	977	7.21	49	2	−1.4	2.8	1.5	1.1	−2.8
Boli	6	4zao6	165	0.56	642	975	7.45	49	2	1.4	3.2	3.7	6.5	1.3
Hailun	1	Haiyu 6	217	0.30	656	983	15.60	49	2.6	5.1	2.5	−2.8	3.9	5.3
Hailun	2	Haiyu 6	222	0.12	631	924	16.48	49	2.6	5.1	2.5	−2.8	4.1	6.1
Hailun	3	Haiyu 6	158	1.15	601	840	15.77	49	4.9	2.5	3.5	−3.5	2.1	8.8
Hailun	4	Haiyu 6	203	0.46	749	951	16.35	49	2.6	5.1	2.5	−2.8	3.7	6.8
Hailun	5	Haiyu 6	163	1.40	925	971	16.12	49	2.6	5.1	2.5	−3.5	3.3	6.9
Hailun	6	Haiyu 6	197	0.64	920	964	16.21	49	2.6	5.1	2.5	−3.5	3.3	7.2
Tonghua	1	Jidan 159	301	0.24	676	827	8.676	49	0.8	0.0	0.5	−2.1	5.8	0.1
Tonghua	2	Jidan 159	281	0.69	698	574	11.46	49	0.8	0.0	2.1	−0.7	5.8	0.2
Tonghua	3	Jidan 159	293	0.52	685	746	9.282	49	0.8	0.0	1.1	−1.4	7.7	0.1
Tonghua	4	Jidan 159	259	1.15	680	701	9.938	49	0.8	0.0	0.7	−2.1	6.9	1.4
Tonghua	5	Jidan 159	267	0.86	686	652	10.5	49	0.8	0.0	1.1	−1.4	5.8	0.1
Tonghua	6	Jidan 159	232	1.73	690	814	8.425	49	0.8	0.0	1.4	−1.4	6.8	1.5
Huadian	1	Limin15	186	1.61	581	970	8.75	49	1.1	−1.1	3.8	5.0	6.9	1.8
Huadian	2	Limin15	228	0.20	590	975	9.454	49	2.5	−3.4	2	0	7	2.1
Huadian	3	Limin15	172	1.59	584	945	9.915	49	2.5	−3.4	2	0	7.1	1.3
Huadian	4	Limin15	217	0.58	604	934	9.242	49	2.5	−3.4	2	0	7.1	1.3
Huadian	5	Limin15	176	1.76	597	949	9.345	49	2.5	−3.4	2	0	7.1	1.5
Huadian	6	Limin15	197	1.22	599	945	9.413	49	2.5	−3.4	2	0	7.1	1.6
Wafangdian	1	Lianyu 6	296	1.21	740	708	11.64	49	1.7	2.3	2.6	−0.7	4.9	−1.0
Wafangdian	2	Lianyu 6	296	1.42	679	978	14.99	49	1.7	2.3	2.1	−5.0	10.8	−7.5
Wafangdian	3	Lianyu 6	293	1	736	818	8.928	49	2.3	0.0	1.1	−3.5	8.5	−5.1
Wafangdian	4	Lianyu 6	308	1.12	707	935	15.72	49	1.7	2.3	1.1	−3.5	8	−4.6
Wafangdian	5	Lianyu 6	296	1.19	715	490	16.23	49	1.7	2.3	1.1	−2.8	6.7	−3.8
Wafangdian	6	Lianyu 6	280	1.53	727	584	13.87	49	1.7	2.3	1.5	−2.1	5.4	−1.5
Dengta	1	Dd60	310	0.77	761	460	16.09	49	1.6	3.6	2	−2.1	10	8.6
Dengta	2	Dd60	322	0.18	779	962	8.295	49	1.6	2.4	2	−2.8	7.3	6.6
Dengta	3	Dd60	311	0.65	780	482	14.25	49	1.6	3.6	1.1	0	9.9	7.5
Dengta	4	Dd60	291	1.07	779	415	16.28	49	2.2	2.4	0.5	−2.8	9.6	3.4
Dengta	5	Dd60	295	1.01	787	437	15.56	49	2.8	3.6	0.5	1.4	9.9	7.8
Dengta	6	Dd60	323	0.09	814	958	7.33	49	1.6	2.4	2	1.4	6.5	4.9

* Dongdan60 is abbreviated Dd60.

## Data Availability

China Meteorological Agricultural Meteorological Experimental Stations (AMESs; http://data.cma.cn/) (accessed on 20 May 2020); Chinese soil scientific database (http://vdb3.soil.csdb.cn) (accessed on 10 June 2020); Institute of Crop Sciences of Chinese Academy of Agricultural Sciences (http://ics.caas.cn/en/) (accessed on 15 February 2020).

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
