# Peer review of "Yield Response of Spring Maize under Future Climate and the Effects of Adaptation Measures in Northeast China"

_plants, 2022, doi:10.3390/plants11131634_

Round 1

Reviewer 1 Report

1. Climate change is a serious problem globally, so the use of a prediction model to simulate the future situations of agricultural-related problems is valuable for the field of food security. In my opinion, this present study is valuable.

2. Abstract: The sentence "In conclusion, for high temperature sensitive varieties, early planting, cultivar change and use of automatic irrigation would generally be an effective adaptation option to reduce yield loss from climate change." is not the main result of this present study. In my opinion, this is a suggestion for maize farmers when facing high temperatures in the future. To avoid misunderstanding, this sentence needs to be rewritten.

3. L13: the regions mostly affected - the regions most affected

4. L16: CERES-Maize model - the CERES-Maize model

5. L20: and decrease in sunshine hours - and a decrease in sunshine hours

6. L23: in reduction in maize yield - in a reduction in maize yield

7. Table 1: to changes in photoperiod - to changes in photoperiod.

8. Table 1:  successive leaf tip appearances -  successive leaf tip appearances.

9. The authors need to enrich the part of the discussion by comprehensively describing the correlated literature on climate change, modeling and crop yield, etc.

Reviewer 2 Report

Dear Authors,

The reviewed manuscript „Uncertainty Analysis of the Impact of Future Climate Change on the Yield of Spring Maize and the Effect of Resilient Measures on it Under Different Parameters in Northeast China” contains the results of an interestingly planned study to assess the uncertainty of the impact of future climatic change on the yield of spring maize in Northeast China and the effect of resilient measures under different parameters for two future periods, relative to the baseline under two different scenarios. For this purpose, the authors adapted CERES-Maize model (DSSAT) v4.7. It was found that that increased temperatures and decrease in sunshine hours and precipitation levels would shorten the maize growth durations and this would result in reduction in maize yield. In conclusion some adaptation measures were proposed to mitigate against the impacts of climate change on spring maize production in region of study.

Both the introduction to the research problem and the project assumptions, and the methodology were presented in a clear and essential manner, sufficient for the needs of the article submitted for review. The results obtained can be used for. However, I have few minor comments on it that should be included to improve its clarity.

L 89 – wrong Figure number

Some figures are not cited in the text (figure 2, 4, 5). Missing figure 16.

Conclusions should be rewritten and shortened.

Reviewer 3 Report

The article is interesting and falls within the scope of the journal. The article presented the results of the study Uncertainty Analysis of the Impact of Future Climate Change on the Yield of Spring Maize and the Effect of Resilient Measures on it Under Different Parameters in Northeast China”. The title of the article and the interpretation of their results are showing good consistency. However, improvements like article write-up, sentence structure and short sentences would make the manuscript more effective.

The abstract of the article looks a little bit short, not well describing the results. The authors need to improve it in order to improve the worth of the paper. The introduction part is well written. However, the authors should improve some sentences shown in specific comments. Certain sentences have no reference cited. Authors should also need to add a paragraph on how better management practices would result in enhanced maize yield and cite their references.

The materials and methods are well presented as well as the description of the results has been done correctly. The presented discussion is correct, based on the objectives of the present research work.

Specific Comments:

Line 63-65- No reference has been given for improving management practices such as irrigation, planting density and fertilizer application.

Line 88- Replace the “in reference to spring maize will be selected from the Agricultural” by the suitable study sites will be selected”

Line 90- Replace “the suitable study sites will be selected” by “the suitable study sites were selected”

Line 101- Author should change the sentence “crop genotype and crop phenotype.” to “crop genotype and phenotype”

Round 2

Reviewer 1 Report

It has been revised accordingly and looks better now.

Author Response

Thank you for your comments and there is no more need to revise. please do take the next step.